# Advanced Insights into Walnut Protein: Structure, Physiochemical Properties and Applications

**DOI:** 10.3390/foods12193603

**Published:** 2023-09-28

**Authors:** Yuxuan Zhao, Weiheng He, Sihan Zhao, Teng Jiao, Haifang Hu, Jingming Li, Lei Zhang, Jiachen Zang

**Affiliations:** 1College of Food Science and Nutritional Engineering, China Agricultural University, Beijing 100083, China; 13722890501@163.com (Y.Z.); heweiheng2022@163.com (W.H.); zshzsh202309@126.com (S.Z.);; 2Academy of Forestry Sciences, Urumqi 830062, China; 3College of Forestry and Landscape Architecture, Xinjiang Agricultural University, Urumqi 830052, China

**Keywords:** walnut, protein, structure, function, sustainability

## Abstract

Facing extreme pressure from an increasing population and climate degeneration, it is important to explore a green, safe and environmentally sustainable food source, especially for protein-enriched diets. Plant proteins have gained much attention in recent years, ascribing to their high nutritional value and environmental friendliness. In this review, we summarized recent advances in walnut protein with respect to its geographical distribution, structural and physiochemical properties and functional attributes. As a worldwide cultivated and largely consumptive crop, allergies and some physicochemical limitations have also led to a few concerns about walnut protein. Through comprehensive analysis and discussion, some strategies may be useful for future research, extraction and processing of walnut protein.

## 1. Introduction

In recent decades, demographic growth and associated climate problems have posed a threat to the sustainability of global food systems. The accelerated expansion of food manufacture triggered great greenhouse gas (GHG) emissions. Therefore, exploiting environmentally friendly food sources, especially alternative protein sources, is a promising way to solve the conflict between global food shortage and environmental challenges. The walnut (*Juglans regia* L.), a member of the Juglandaceae plant family, is one of the most widespread tree nuts throughout the world owing to its nutritional value. Being authorized by the FDA (Food and Drug Administration) as a nutritional diet, which may reduce the risk of various diseases, walnuts are often used as ingredients in versatile foodstuffs to enhance their nutritive content [1]. Some previous studies have proved that the oil content in walnut is up to 62–68%, containing a large amount of polyunsaturated fatty acids. Thus, except for a minor percentage of harvested walnut consumed freshly, a considerable amount of walnuts are processed to be oil.

However, in the process of walnut production, a large amount of walnut meal is produced, which wastes resources and pollutes the environment. More than 52% of the total mass of walnut meal is walnut protein, but it is usually underutilized or used as animal feed due to its poor functional properties such as solubility [2]. If walnut meal protein resources can be comprehensively utilized and their application scope broadened, it will be of great significance to the sustainable development of the walnut industry and the development of health food [3].

The aim of this article is to provide a systematic review on walnut protein, with a focus on its physicochemical properties. Owing to its promising roles in food sustainability, the global distribution of walnuts was displayed and compared. To further facilitate the utilization of walnut protein for human health, its nutritional attributes were also discussed. In regard to current concerns of walnut protein applications, we put forward some proper strategies, which may also be useful for the exploration of emerging plant protein sources.

## 2. Global Distribution and Characteristics of Different Species

According to the data in 2018, an area of 1.186.398 ha is allocated for walnut cultivation, with an annual production of over 3.6 MT [4], in which half of the walnuts are produced in China, followed by the USA, Iran, and Turkey [5]. The typical walnut cultivating countries distributed around the world are summarized in Figure 1, according to their production amount. The content and quality of protein in walnuts is dependent on the species and environmental factors, including season, temperature, moisture and light. Additionally, the distribution is also closely related to the population and historical tradition. Walnut belongs to the Juglandaceae family (*Juglans*) and there are more than 20 species of walnut, distributed across Asia, Europe and the Americas (mainly North America).

### 2.1. Asia

As the leading country in terms of global walnut production, there are more than 20 typical species of walnut widely cultivated in China. The average protein content of *Juglans sigillata*, mainly cultivated in Yunnan Province, is about 15%. Following Yunnan province, Shanxi is the second largest province for walnut production. Among the mainly locally cultivated Fenyang walnuts, the fat content is more than 57% and the protein content is more than 15%. In addition to the above provinces, Xinjiang is suitable for walnut growth because of its long daylight time and the large temperature difference between day and night. The average protein content of the Aksu walnut cultivated in Xinjiang is 19.35%, and the essential amino acid content accounts for 27~30% of the total amino acid content, which meets the healthy demand of human beings.

Iran, with 9.1% of the world’s walnut production, is the third leading walnut producer in the world [6]. Also, considering the harvested area, Iran is the fifth leading country, occupying 4.9% of the worldwide walnut orchards [7]. In addition, the cloned varieties of walnut are also of great interest in the modern market, such as the “Qingxiang” walnut, which is sourced from Nagano, Japan. The average protein content within the nuts is around 23.1%. The outstanding flavour, quality and disease-resistance have pushed forward this advanced species broadly grafted into other countries.

### 2.2. Europe

Walnuts are cultivated throughout Turkey with a planting area of 41,393.2 ha. As shown in Figure 1, the Aegean region has the largest cultivated area of 7696.3 ha, followed by the Anatolia region with a cultivation area of 5678.9 ha. The main varieties, including Yalova, Sebin, Bilecik, Kaplan, Gultekin and Sen, have a protein content above 16%, among which the content in Yalova No. 1 is as high as 23% [8].

Walnuts, grown wild or semi-wild, covered almost all of Romania and accounted for 3% of all fruit production in the decade from 2000 to 2010, according to Romania’s 2011 Agro-Forestry Statistical Yearbook [9]. Up to the end of 2016, the most cultivated walnuts were Jupâneşti, Mihaela, Roxana, Valcor, Valrex, Valmit, Geoagiu65, Velniţa and Miroslava. Franquette, originating in France, has been introduced and cultivated in a large number of walnut producing areas in Europe and the United States. It is a late-fruiting walnut variety with an average fruit weight of 11.09 g and a yield of 46%, endowed with economic advantages.

### 2.3. America

By 2010, the national walnut cultivation area was 107,600 ha in the United States, and the annual output of nuts was 525,300 t, of which 99% was concentrated in California. The main cultivated varieties are Chandler, Hartley, Serr, Howard, Vina, Payne, Ashley, etc. Among the varieties, Chandler accounts for 37% of the walnut acreage in the United States, and Chandler has the leading yield with great protein potential [10].

## 3. Structural and Physiochemical Properties of Walnut Proteins

### 3.1. Composition and Molecular Structure of Walnut Protein

Naturally existing proteins have a well-defined tertiary structure owing to their polypeptide sequences and resulting molecular forces, including electrostatic forces, ion pairing, van der Waals interactions, hydrogen bonds and hydrophobic effects. The dynamic conformation of each protein influences its physiochemical properties and nutritional function in food. Based on their intrinsically different structures, walnut proteins can be divided into four types of protein, namely water-soluble albumin, salt-soluble globulin, alcohol-soluble prolamins and alkaline-soluble glutens, occupying 7.54%, 15.67%, 4.73% and 72.06% of the total amount of walnut protein, respectively.

Gluten, the primary storage protein of walnuts, accounting for the major portion of walnut proteins, includes gliadins and glutenins owning to their solubility in aqueous alcohol [11]. Glutenins could polymerize disparate subunits to form glutenin macropolymer by inter-chain and intra-chain disulphide bonds [12]. While gliadins are monomeric proteins stabilized by intra-chain disulphide bonds, which can decrease the high elasticity of glutenins [13,14]. More than four major polypeptides have been identified with the albumin and globulin fractions, ranging from 12,000 to 21,000 kDa. Additionally, Kar, Sze-Tao, and Shridhar [15] estimated several additional high-molecular-weight polypeptides from albumin and globulin, ranging from 25,000 to 85,000 kDa, respectively.

Mao et al. [16] analysed the amino acid composition of walnut protein and four independent proteins, proving that gluten, albumin and globulin have a balanced content of essential amino acids, meeting the FAO’s recommendation for adults (Table 1). Comparably, walnut protein and the four independent proteins are rich in Glu and Arg, two typical hydrophilic amino acids. The distribution of these amino acids among four fractions is uniform. While there are significant differences between the water solubility of the four proteins, which may be caused by their tertiary structure. The isoelectric point of the walnut protein component ranges from pH 4.8 to 6.8 [16]. An analysis using circular binary chromatography (CD) shows that the secondary structure of walnut protein mainly includes an α-helix, β-fold, β-rotation and random coil, among which the α-helix and random coil are the most common [17].

To better utilize the walnut proteins and their fractions, the understanding of their high structure is of key significance, which was rarely reported up to now. 7S vicilin, a kind of typical globulins, is one of the rare walnut proteins reported with a crystal structure. According to the Protein Data Bank (PDB) information from Zhang [18], the walnut 7S was observed as two trimeric biological units in one asymmetric unit (Figure 2a). Moreover, the walnut vicilin is the only copper-binding protein reported so far, compared to vicilin from other legumes or nuts. To obtain more details of walnut proteins without a crystal structure reported, we predicted the tertiary structure of 2S albumin by Rosetta software (a comprehensive set of software for simulating macromolecular structures), according to the sequence reported from Suzanne and Guo and Zhang. As shown in Figure 2b, α-helix domains are linked by loops, among which three disulphide bonds assist with stabilizing the structure. The energy calculation of amino acids in Figure 2c indicates that the low energy score contributes to the high water solubility of 2S albumin. In regard to gluten and prolamin, their structures exhibit high heterogeneity owing to their complex composition and low solubility. We obtained a partial sequence of walnut glutelin after searching the National Center for Biotechnology Information (NCBI), and further predicted its high structure by Rosetta. The structure with the highest estimated score, up to 0.938, was drawn in Figure 2c. Compared to the other two proteins, there are more fractions of the β-sheet structure located in the predicted partial structure of glutelin, resulting in the much higher energy score calculated in Figure 2e, which may be a key factor leading to its low solubility.

### 3.2. Properties of Walnut Protein

Protein properties (such as solubility and emulsification, as well as the ability to hold water and oil) are the basis for their specific functions in food processing. At present, most of the studies on the functional properties of walnut protein are issued at the level of total walnut protein.

#### 3.2.1. Solubility

Solubility is an important parameter to evaluate the quality of protein, and also the premise for protein to play other functional roles. It is mainly affected by solvent, temperature, pH value and other conditions. In general, the pH dependence of walnut protein solubility is consistent with trends observed for many plant proteins [19,20,21]. When the pH value is 4, the soluble protein content is only 2.17%, while when the pH value is greater than 8, the maximum solubility is about 90% [15]. Being sensitive to heat, the solubility of walnut protein reaches 55% at 55 °C, and then decreases with the increase in temperature until the denaturation temperature of 67.05 °C [22]. A protein with good solubility can promote its application in the production of protein drinks. Therefore, scholars are constantly exploring and improving the extraction process of walnut protein.

Protein solubility is closely related to the balance between surface hydrophobicity and surface charge. Aggregation of protein molecules can be induced by surface hydrophobicity, whereas surface charges help inhibit aggregation [23]. Among all the solvents in Sze-Tao’s work, 0.1 M NaOH had the best solubilization effect, while 70% ethanol aqueous solution had the lowest protein mass [15]. Kong et al. proved that high-speed shearing was useful for improving the solubility of walnut protein, especially at an alkaline pH [24].

High-intensity ultrasound and high-pressure treatment has proven to be effective among various proteins with low solubility, such as whey proteins and casein [25]. In addition to protein yield and properties, turbidity, viscosity, gelation point and the elasticity of protein gels may also be changed by the above treatment [26]. Zhu et al. reported that ultrasonic treatment increased the water solubility of aqueous suspensions of walnut protein by 22% [3].

Besides the physical modification methods, chemical modification is also popular in changing the intrinsic structure and surface characteristics of food proteins, improving their solubility and functional properties [27,28]. For instance, sodium trimetaphosphate (STMP) and sodium tripolyphosphate (STP) have been broadly applied in modifying the solubility and emulsifying properties of plant proteins [29]. The application of enzymes in walnut protein extraction can not only change its physical and chemical properties and functional properties, but also increase its digestive and absorbing ability. More detailed information on increasing the water solubility of walnut protein will be discussed below.

#### 3.2.2. Emulsification

Deng et al. have studied the physical, chemical and functional properties of walnut proteins and their main components. The emulsification stability of globulin is the highest among all walnut proteins, which is helpful for its use as an emulsifier in food processing [30]. There is a positive correlation between emulsification and solubility, which is affected by pH and salt ion concentration, as well as its own concentration and external temperature. Qin et al. presented that high hydrostatic pressure is capable of increasing the emulsion activity, but decreases the emulsion stability of the walnut protein isolate. The solubility was not increased, while in vivo digestibility was apparently enhanced [31].

Furthermore, ultrasound is used as a promising emulsification technique to prepare protein emulsions [32,33]. Solvents and extraction time can be significantly reduced in ultrasonic-assisted extraction (UAE) rather than in traditional methods [34]. During the treatment, cell structures can be effectively disrupted by strong shear forces from ultrasound, with an improvement of the mass and heat transfer [35,36], raising the emulsification rate. The results were in accordance with the reports that long-time and high-power ultrasound could improve the emulsifying properties of peanut proteins and walnut protein isolates, possibly because the loosing structure of the protein under ultrasonic treatment would cause better adsorption of the protein at the oil–water interface.

Besides protein, walnut meal contains residual lipids as well, which is easily oxidized along with protein oxidation in protein-based emulsions. Co-oxidation of lipid and protein always occurs at the oil–water interface [37], which seriously affects food emulsions’ stability [38]. Polyphenols, including catechin, chlorogenic acid, anthocyanin and resveratrol, have been demonstrated to improve proteins’ emulsifying attributes via non-covalent interactions (e.g., hydrogen bonds and hydrophobic interactions) or covalent interactions [39,40,41]. Therein, the incorporation of protein and polyphenols obstruct the interaction between protein and lipid, depressing the emulsion aggregation [42]. Simultaneously, polyphenols have been reported to possess an excellent antioxidant ability, serving as free-radical scavengers at the oil–water interface.

#### 3.2.3. Water and Oil Retention

Water retention is the ability of hydrated protein to hold water firmly without loss. At present, soybean protein isolates and soybean protein concentrates are widely used in meat products, mainly to improve the quality and yield of products by using their characteristics of good water retention. Not only intrinsic factors, including amino acid composition, protein conformation and surface polarity, but also extrinsic parameters, like pH, temperature and ionic strength, play key roles in affecting the water-holding ability of walnut protein. Therefore, the study of the holding power of walnut protein has certain significance for the application of walnut protein.

Hu et al. analysed the effects of pH and salt concentration on water and oil retention of walnut protein [43]. They found that at the isoelectric point of walnut protein, the interaction between proteins was the largest, while the water-holding capacity of walnut protein was the lowest. When pH is adjusted to above 4, the water-holding ability increased significantly. After ionization, the interaction between proteins decreased, but the water-holding capacity was stronger [44]. In 0–0.6 mol/L NaCl solution, the water-holding capacity of walnut protein increased with the increase in ion concentration. At a low salt concentration, water molecules around proteins can be reduced owing to the combination of charged protein molecules and hydrated salt ions. Nevertheless, at a high salt concentration, this led to the dehydration of the protein, by improving the interaction between water and salt ions [20]. When the temperature was 40 °C, the maximum water-holding capacity of walnut protein was 3.55, much higher than before. When it is lower than or higher than this temperature, the hydraulic capacity decreases, and at a high temperature, it may cause a reduction in the hydrogen bond strength and protein denaturation.

Hu et al. also compared the oil-holding capacity of walnut protein obtained by a different method, concluding that the oil retention of protein from enzyme-assisted reverse micelles was obviously higher than from other methods [43]. The most likely reason is that the enzyme could hydrolyse walnut protein, leading to the exposure of hydrophobic amino acids, which prefer to bind to more aliphatic hydrocarbon chains. In addition, the structure change during protein processing is also an important factor for the changing of the oil-holding capacity. The gluten in walnut cake protein was modified using the enzymatic method and the microstructure of the walnut protein was changed, thereby improving the functional properties of the walnut protein. Sun et al. found that the modified walnut glutelin (WG) increased the solubility by ~1.33 times, the water holding capacity by ~0.23 times, the emulsifiability by ~0.32 times, and the emulsion stability by ~0.75 times. The oiliness was slightly lowered and the foaming characteristics were not greatly changed [45].

## 4. Functional Attributes of Walnut Proteins

Based on its high nutritional and functional properties, such as anticancer, anti-inflammation, antioxidant activity and several pharmacological activities like brain and cardiovascular protection, diabetes treatment, etc., the walnut has always been outstanding among various functional foods [46]. Except for lipid nutritive, the walnut is a healthy source of protein, which is rich in essential amino acids, close to the FAO and the WHO standards [15].

### 4.1. Walnut Peptide

Bioactive peptides are proteins with a different composition and arrangement from 2 to 20 amino acid residues, consisting of linear and circular structures [47]. Bioactive peptides are superior to free amino acids and proteins in physiological function and nutritional significance. Along with the rapid and broadened development of walnut cultivation and processing, more and more bioactive peptides identified from walnut proteins are becoming attractive.

These compounds have a small molecular weight, which can be directly absorbed and utilized by the human body without digestion, and have a variety of biological activities, especially protective effects against neuroinflammation and oxidative stress. For example, the tripeptide Leu-Pro-Phe (LPF) shows strong anti-neuroinflammatory effects, owing to its high hydrophobicity, facilitating peptide passage through cell membranes [48]. It may be caused that leucine or proline-rich peptides exhibit anti-inflammatory activity by inhibiting the TLR4-MyD88 signalling pathway or NF-κB activation [49]. Moreover, some investigators have reported that walnut peptides can improve oxidative stress and reduce the neuroinflammation caused by excessive proinflammatory factors such as neurodegenerative diseases [50]. Gu et al. [51] isolated walnut peptides with free radical scavenging and PC12 cytoprotective activities; Jahanbani et al. [52] reported a direct relationship between antioxidant activity and anticancer activity of walnut peptides. Zou et al. [53] suggested that walnut peptides may protect against Alzheimer’s disease by regulating the antioxidant system. Degreased walnut meal hydrolysate can protect PC12 cells from H_2_O_2_-induced oxidative damage and improve memory impairment in mice [54]. EVSGPGLSPN (Glu-Val-Ser-Gly-Pro-Gly-Leu-Ser-Pro-Asn) can alleviate H_2_O_2_-induced neuroinflammation by increasing the activity of antioxidant enzymes and inhibiting the over-activation of NF-κB and caspase [55]. Owing to their immunomodulatory effects and antioxidant activity, walnut peptides can also regulate lipid metabolism.

Moreover, Liu et al. purified and identified several ACE (angiotensin l-converting enzyme) inhibiting peptides from walnut protein [56]. Cong et al. further analysed the interaction of ACE using the molecular docking method based on ligand–receptor locking principle [57]. The fragment spectra of walnut hydrolysate were analysed by mass spectrometry [58], identifying 33 unique peptides. In general, peptides obtained from walnut protein are becoming recognized ingredients not only in food supplementation, but also in clinical therapy. A more detailed classification of hydrolysate or peptides from walnut proteins is summarized in Figure 3, based on their functionality.

Considering the manufacture of peptides, enzymatic hydrolysis, fermentation and in vivo or in vitro digestion are the main methods to release peptides from proteins. Neutral protease, like alkaline protease, flavour protease, papain, trypsin, pepsin and compound protease have been reported to be used in the enzymatic hydrolyzation of walnut protein. In view of the advantages of specificity, efficiency and gentleness, enzymatic hydrolysis has become the most widely used and mature preparation method. In Wu’s report, they established enzymatic membrane reactors, equipped with a gradient diafiltration feeding technique, for the continuous production of multi-functional peptides [59]. To increase hydrolysis efficiency, physical assistance methods including ultrasound, high pressure and microwaves have been applied [60].

### 4.2. Encapsulation

Plant proteins are preferred to animal proteins and other synthetic polymers as nanocarriers due to their safety and good stability in vivo [61]. They also overcome the risk of animal proteins that may carry pathogens [62]. As a plant protein with great potential, walnut protein can naturally absorb non-polar molecules such as fat [63] and has good emulsification and solubility under alkaline conditions [15]. The development of microcapsules with lipid-soluble active components using walnut protein as a wall material can significantly improve the solubility and stability of the coated material, which provides a new direction for the application of walnut protein.

Curcumin (CCM) is a natural lipophilic polyphenol in the rhizome of turmeric, which promotes a variety of pharmacological activities, such as antioxidant, anticancer and anti-inflammatory agents. However, the low water solubility and poor bioavailability of CCM greatly limit its application in functional food and nutritional supplement formulations. In Asadi and co-workers’ report, CCM was successfully encapsulated in a complex nanoparticle composed of walnut proteins (WP NPs) with the encapsulation efficiency reaching 61.5% [63]. The addition of curcumin affected the size, form and surface properties of the electrosprayed walnut protein nanoparticles. The CCM can stay intact in the stomach due to the protective effect of the walnut protein, while release in the intestine with the hydrolyzation of walnut protein nanoparticles, enhances its bioavailability [64]. Furthermore, the antioxidant activity of CCM-WP NPs after in vitro digestion was significantly higher than in WP NPs due to the bioactive peptides released from WP NPs. Moghadam et al. have established a pH-shifting method to encapsulate CCM into walnut-protein-employed nano carriers. The complexed samples exhibited significant in vitro bioactivities against the proliferation of metastasis and non-metastasis human breast cancer cell lines [65]. To develop the adhesion of natural plant proteins, Lei et al. investigated the ethanol-treated complex of walnut protein and xanthan gum, gaining much better rheological and texture properties [66]. Therefore, it can be concluded that the PT-WNPs complex may be an effective delivery system for bioactive compounds.

In our recent work, walnut protein and gum Arabic were complexed to construct a nanocarrier for carvacrol, a natural phenolic compound with excellent antibacterial and antioxidant activities. As show in Figure 4, the optimum preparation conditions were achieved at pH 4.0 with a protein-to-gum Arabic ratio of 6:1 (*w*/*w*). The microencapsulation of carvacrol not only improved its thermal stability, but also facilitate its intestinal absorption by resisting gastric acid. Moreover, the microcapsules presented antibacterial activities, indicating the potential value of the walnut in the area of food preservation [67].

### 4.3. Product

As a functional food, walnuts have attracted much attention in promoting health, and a diet rich in walnuts can reduce the levels of inflammatory factors in the body [68]. The general processing of raw walnuts includes cleaning, soaking, peeling and colloid grinding (Figure 4). Following some specific strategies to modify the protein characteristics, typical walnut products, including drinks and additives, can be obtained. Besides the products mentioned in Figure 4, there are many other products containing nuts, such as bread, meat products, etc.

Traditional walnut milk is a popular and nutritional drink, a good source of protein, vitamin B, nicotinic acid and a variety of trace elements, and has a strong walnut fragrance. Gharibzahedi et al. conducted a response surface modelling experiment to optimize the manufacturing conditions. They concluded that the water/oil ratio was the most important factor, influencing turbidity loss rate, size index and opacity [69]. Liu and colleagues investigated the impact of pH, freeze–thaw and thermal sterilization on the physicochemical stability of a walnut beverage. The mixed emulsifier had a significant effect on the physical stability, while a longer thermal sterilization induced a poorer physical and oxidative stability of the beverage, which may destruct the interfacial layer of walnut protein and xanthan gum [70]. Fermentation is also a popular processing method in the manufacture of walnut beverages, which may increase the taste and nutritional attributes of walnuts. Cui applied kefir grains, as an inoculum, in the preparation of a walnut milk beverage, with the suggested optimum conditions as follows to maximumly develop its sensory quality: a temperature of 30 °C, time of 12 h, inoculum size of 3 g and sucrose concentration of 8 g/100 mL [71]. Considering food safety problems in plant-sourced beverages, Ding et al. designed and established a strategy, based on DNA barcoding, which can be used to identify raw ingredients in walnut milk beverages [72].

As a good adhesive, filler and flavour enhancer, walnut protein has been widely added to meat products such as sausage and ham, maintaining products exquisite organization. Given the unique combination of nutrients and phytochemicals in walnuts, Serrano et al. evaluated the physiochemical and sensory characteristics of restructured beef steak supplemented with different proportions of walnuts. The morphology characteristics suggest that walnuts interfere with the formation of protein network structures, presenting more acceptable properties [73]. In Serrano’s work, they found that supplementation of walnuts could significantly increase the frozen stability of restructured beef steak, without any adverse effects [74]. Thus, incorporation of nuts in meat products can be used to confer potential heart health benefits. In addition, the consumption of walnut-rich meat increased the antioxidant status of volunteers, who were at risk of coronary heart disease [75]. Therefore, the walnut may be a promising raw material for functional food, for it can supply adequate energy for overweight and obese people without negative effects on body weight.

## 5. Concerns and Discussion

### 5.1. Allergy

Tree nut allergies affect an estimated 0.6% of the population of the United States [76], with the English walnut (*Juglans regia*) as one of the most commonly reported allergenic tree nuts [77]. Severe and life-threatening reactions from walnut proteins have been reported in numerous surveys [78,79,80].

Four major allergens have been documented in English walnuts: Jug r 1, a 2S albumin [81]; Jug r 2, a 7S vicilin-like seed storage globulin [82]; Jug r 3, a non-specific lipid transfer protein [83]; and Jug r 4, an 11S legumin-like seed storage globulin [84]. Specific information is shown in Table 2. Sordet et al. evaluated the binding site of Jug r 1 to human serum antibodies by using overlapping peptide library screening, identifying four epitopes. These linear IgE epitopes have a flexible hypervariable region on the surface of the solvent, which can bind to the specific IgE of the patient serum and produce sensitization [85]. Barre et al. [86] reported IgE-binding epitopes on the surface of three-dimensional models of pea globulin in the walnut (Jug r 2), hazelnut (Cor a 11), cashew (Anacardium occidentalie) (Ana o 1) and peanut (Ara h 1). Jug r 2 and Ara h 1 of the peanut allergen showed high homology in sequence and spatial structure, but the cross-reactivity between them was low in vitro. These proteins are thermally stable and can maintain their spatial structure at temperatures below 75 °C [87]. Experiments have shown that the sensitization of Jug r 2 does not change significantly after food processing methods such as arm-irradiation, microwave, frying and grilling [88]. Therefore, vicilin (Jug r 2) is also considered as a class I food allergen. Jiang et al. found that the dry heat treatment had no significant effect on the IgE binding capacity and IgG binding capacity of walnut protein, while wet heat treatment significantly reduced the IgE and IgG binding capacities of walnut protein. Compared with dry heat treatment, wet heat treatment can significantly reduce the antigenicity of walnut protein. This may be due to the different heating and heat treatment modes of the two heating methods, and the different water content, resulting in different rates of certain chemical reactions (hydrophobic and covalent) of protein molecules, thus affecting its antigenicity. Moist heat treatment is more likely to denature and coagulate protein and destroy the protein chemical structure, thus destroying the allergen epitope and reducing protein sensitization [89].

It is difficult to detect the allergenic protein because it is low in the walnut and often coated with a food matrix. Moreover, the decreased solubility of walnut protein may affect most results for detecting and quantifying the presence of allergenic food residues [12]. Thus, in order to ensure the safety of consumers and strengthen the management of allergens, it is urgent to develop and adopt an efficient and sensitive method for detecting walnut allergens [90].

### 5.2. Physicochemical Limitations

The main protein group (about 70%) of walnut is glutenin, and its poor water solubility limits its functional properties as a water-based food. Therefore, it is important to improve the physicochemical properties of walnut protein through various ways.

As a traditional method, the alkali solution extraction method has been widely used in protein extraction due to its advantages of simple operation and economic efficiency [60]. However, alkali solution extraction has the disadvantages of large solvent consumption, low extraction efficiency and high energy consumption. Therefore, exploring a green, safe and low-cost protein extraction technology is the key to meet the needs of high extraction efficiency and environmental protection.

High-intensity ultrasound (HIU) is an emerging non-thermal technology in the food industry, which has the advantages of a high extraction rate, less solvent consumption, low maintenance cost and environmental sustainability. A large number of studies have shown that ultrasound-assisted protein extraction has powerful properties in improving protein yield and changing physical, structural and functional properties of proteins, such as whey protein [91], soy protein [92] and meat protein [93]. This is mainly due to the combination of physical and chemical interactions caused by acoustic cavitation, which further enhances mass transfer during the processing. High-intensity ultrasound breaks some of the physical forces that hold protein molecules together, thereby releasing smaller soluble proteins [94]. In addition, the ultrasonic treatment may cause some structural changes in individual protein molecules that alter their conformation and thus their water solubility [95]. Ultrasonic treatment increased the water solubility of walnut protein, reduced the number of large aggregates and enhanced the functional properties of walnut protein. However, ultrasound conditions, such as power level and duration, need to be optimized. These results may have important implications for increasing the utilization of walnut protein as a natural plant functional component in food and beverage proteins.

In addition, reverse micellar extraction (RME) is also considered to be an ideal alternative to traditional methods, allowing the simultaneous separation of oils and proteins by reverse micellar systems [96]. Reverse micelles (RMs) are formed by surfactants dissolved in organic solvents and have nanometre-sized aggregates that contain an internal water core that can trap some biomolecules, such as proteins and enzymes [96,97]. This method has a low energy consumption, low possibility of protein denaturation, stable thermodynamics, good interface performance, is environmentally friendly and there is no need to use a large number of acid-base solutions [65]. RME is accomplished by the following two steps: (1) dissolving the biomolecules in the reverse micelle system to obtain the forward extraction; (2) further reverse extraction was obtained by transferring the proteins in the forward extract to the aqueous phase.

## 6. Conclusions and Prospects

This review has focused on the proteins from walnut, which are an emerging alternative source of dietary proteins. We summarized and compared some typical walnut species from Asia, Europe and America, with protein quality emphasized. Water solubility, emulsifying ability, and water and oil-holding abilities are common indecies for protein evaluation. Tertiary structural information of walnut proteins, the prerequisite for their physicochemical characteristics, has been rarely reported. In the future, we may pay more attention to structure determination technologies and molecular docking by artificial intelligence, which may be effective for solving unclear structures. Comparing the properties of different types of walnut proteins with other proteins at different pH levels, salt concentrations or temperatures is also a way to explore the use of plant proteins in different contexts. Walnut peptides have attracted a great deal of attention, ascribing to their diverse functions and enhanced absorption. In addition, walnuts can also be applied as nanocarriers to load some functional materials with poor stability. A few kinds of walnut-protein-sourced products were listed in the manuscript, and the common processing schemes were summarized. Considering the drawbacks of using walnut protein, allergy tops the list for its threat to a great population. Exact identification and proper processing techniques may be effective approaches for allergy control. Finally, physicochemical limitations are an open problem for almost all plant proteins. Besides the strategies mentioned above, exploring a green, safe, and cost-effective protein extraction technique is of great importance, not only for modifying insoluble proteins, but also to enrich the consumable products of walnut protein. Consequently, extensive research and broader development are still required for the walnut, a promising alternative crop with a high content of valuable protein.

## Figures and Tables

**Figure 1 foods-12-03603-f001:**
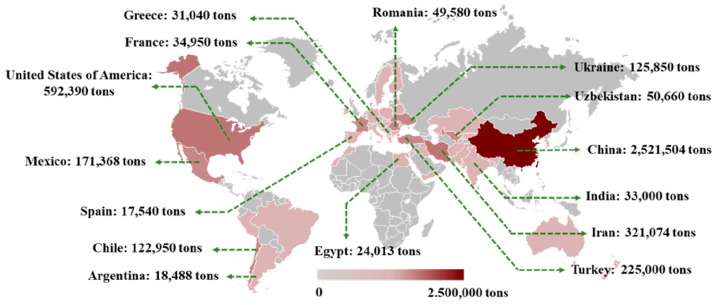
Geographical distribution of walnut cultivation at the country scale, with the annual production identified in 2018.

**Figure 2 foods-12-03603-f002:**
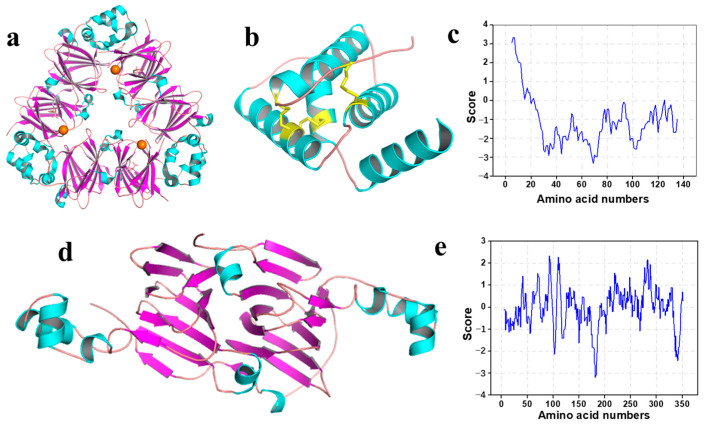
(**a**) The crystal structure of 7S vicilin in walnut, with copper ions presented in sphere. (**b**) The predicted structure of 2S albumin in walnut, and its solubility calculation (**c**). (**d**) The predicted structure of partial glutelin in walnut, and its solubility calculation (**e**).

**Figure 3 foods-12-03603-f003:**
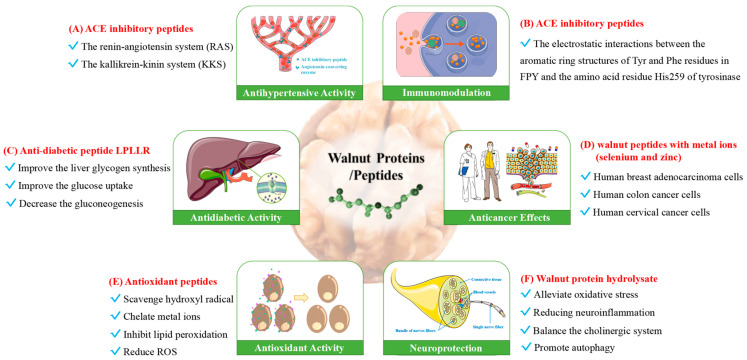
Schematic diagram of multi-functions of walnut protein hydrolysate and peptides.

**Figure 4 foods-12-03603-f004:**
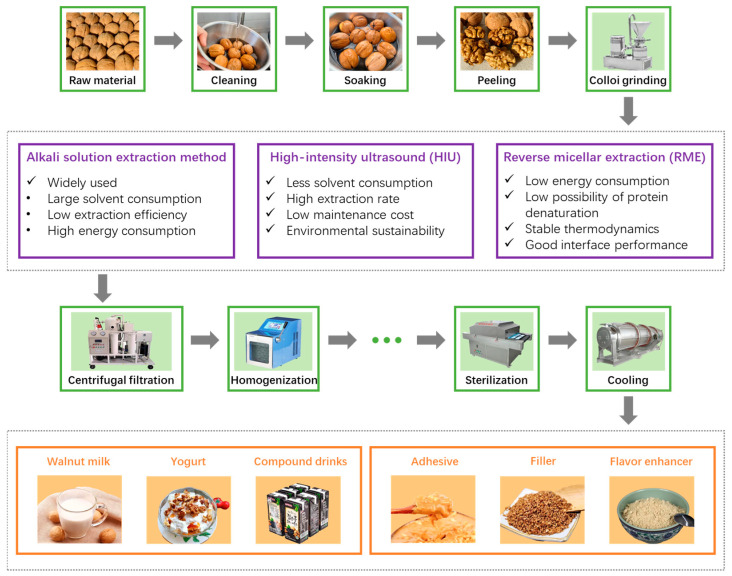
Schematic representation of walnut protein processing and products.

**Table 1 foods-12-03603-t001:** Amino acid composition of walnut proteins and protein fractions (g/100 g) [16].

Amino Acids		Walnut Protein	Albumin	Globulin	Prolamin	Glutelin	FAO/WHO (1990)
Acidic amino acid	Asp	10.04 ± 0.43	8.02 ± 0.57	7.13 ± 0.51	18.08 ± 0.42	10.51 ± 0.44	
	Glu	22.16 ± 0.4	28.7 ± 3.36	28.8 ± 1.26	33.03 ± 1.06	22.7 ± 2.05	
Alkaline amino acid	Arg	14.73 ± 0.42	15.67 ± 0.34	16.01 ± 0.33	17.52 ± 0.43	13.47 ± 0.33	
	Lys	2.58 ± 0.12	3.31 ± 0.16	2.52 ± 0.16	0.83 ± 0.10	1.7 ± 0.17	5.8 (1.6)
Aromatic amino acid	Phe	4.94 ± 0.23	3.89 ± 0.15	3.78 ± 0.08	1.92 ± 0.10	5.11 ± 0.11	6.3 (1.9)
	His	2.38 ± 0.26	2.23 ± 0.14	2.01 ± 0.05	1.4 ± 0.35	2.19 ± 0.16	1.9 (1.6)
	Tyr	2.76 ± 0.11	2.53 ± 0.06	0.76 ± 0.07	3.72 ± 0.09	2.83 ± 0.09	
	Pro	4.22 ± 0.29	4.03 ± 0.10	4.27 ± 0.13	1.64 ± 0.11	5.3 ± 0.24	
Sulfur-containing amino acid	Met	1.16 ± 0.12	1.7 ± 0.10	2.32 ± 0.08	0.84 ± 0.14	1.55 ± 0.11	2.5 (1.7)
	Cys	0.84 ± 0.08	2.21 ± 0.10	1.97 ± 0.09	2 ± 0.04	0.56 ± 0.09	
Neutral amino acid	Gly	5.43 ± 0.07	5.89 ± 0.17	8.73 ± 0.17	7.68 ± 0.27	5.28 ± 0.25	
	Ala	4.74 ± 0.19	3.29 ± 0.24	2.62 ± 0.34	2.57 ± 0.18	4.73 ± 0.27	
	Val	4.18 ± 0.14	3.24 ± 0.11	4.05 ± 0.16	1.49 ± 0.16	4.15 ± 0.16	3.5 (1.3)
	Leu	7.13 ± 0.12	5.21 ± 0.11	5.48 ± 0.16	1.51 ± 0.13	7.31 ± 0.26	6.6 (1.9)
	IIe	3.28 ± 0.15	2.66 ± 0.16	2.79 ± 0.13	0.94 ± 0.07	3.32 ± 0.17	2.8 (1.3)
Hydroxy amino acid	Ser	5.84 ± 0.12	4.8 ± 0.36	5.75 ± 0.23	3.22 ± 0.12	5.81 ± 0.20	
	Thr	3.58 ± 0.20	2.64 ± 0.07	2.02 ± 0.07	1.59 ± 0.13	3.49 ± 0.04	3.4 (0.9)

**Table 2 foods-12-03603-t002:** Identification of common walnut allergens.

Allergen	Molecular Weight/kD	Subtype	Biochemical Classification	Nucleotide Sequence (NCBI)	Protein Sequence (NCBI)	Protein Sequence (UniProt)
Jug r 1	16.4	Jug r 1.01	2S albumin (Prolamin superfamily)	U66866.1	AAB41308.1	P93198
Jug r 2	44	Jug r 2.01	7S globulin (Cupin superfamily)	AF066055.1	AAF18269.1	Q9SEW4
Jug r 3	11.8	Jug r 3.01	Nonspecific lipid transfer Protein (nsLTP) (Prolamin superfamily)	EU780670.1	ACI47547.1	C5H617
Jug r 4	58	Jug r 4.01	11S globulin (Cupin superfamily)	AY692446.1	AAW29810.1	Q2TPW5
Jug n 1	18.9	Jug n 1.01	2S albumin (Prolamin superfamily)	AY102930.1	AAW29810.1	Q7Y1C2
Jug n 2	55.7	Jug n 2.01	Vicilin (Cupin superfamily)	AY102931.1	AAM54366.1	Q7Y1C1

## Data Availability

Not applicable.

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
