# Peer review of "Advanced Insights into Walnut Protein: Structure, Physiochemical Properties and Applications"

_foods, 2023, doi:10.3390/foods12193603_

Round 1

Reviewer 1 Report

The manuscript is well written and presents the key results for walnut proteins. However, in section 2.3 only one work from 2010 is mentioned and the section is very brief in comparison to other regions of the world. It is suggested to expand the information on Latin American countries such as Chile and Mexico with more recent literature from 2017-2023. Also, in section 3.2.1 on solubility, the first paragraph is confusing because the information contradicts that pH 4 presents tremendous solubility, but actually most protein solubility occurs at pH 8. Finally, walnuts present a large amount of allergens and it was not clear if hydrolyzing the proteins increases or decreases their ability to induce an inflammatory response, although the biopeptides of these proteins present different pharmacological targets of interest.

Also, the spelling and writing of some sentences or words from the abstract to the conclusions need to be improved.

Author Response

Response to comments from reviewer 1:

  1. In section 2.3 only one work from 2010 is mentioned and the section is very brief in comparison to other regions of the world. It is suggested to expand the information on Latin American countries such as Chile and Mexico with more recent literature from 2017-2023.

Response: Thanks for your suggestion. We are also investigating walnut varieties and regions in Latin America, but we have not found enough authoritative official data, which may be a state secret or used for some kind of commercial practice.

  1. In section 3.2.1 on solubility, the first paragraph is confusing because the information contradicts that pH 4 presents tremendous solubility, but actually most protein solubility occurs at pH 8.

Response: Suggestion was followed. We have rewritten this sentence as shown in line 177. When the pH value is 4, the soluble protein content is only 2.17%, while when the pH value is greater than 8, the maximum solubility is about 90%.

  1. Walnuts present a large amount of allergens and it was not clear if hydrolyzing the proteins increases or decreases their ability to induce an inflammatory response, although the biopeptides of these proteins present different pharmacological targets of interest.

Response: Thanks for your question. We supplemented some strategies to reduce allergy of walnut proteins from line 434 to 443. Hydrolyzation may be a possible mechanism to reduce inflammatory response. Jiang found that dry heat treatment had no significant effect on IgE binding capacity and IgG binding capacity of walnut protein, while wet heat treatment significantly reduced IgE and IgG binding capacity of walnut protein. This may be due to the different heating and heat treatment modes of the two heating methods, and the different water content, resulting in different rates of certain chemical reactions (hydrophobic, covalent) of protein molecules, thus affecting its antigenicity. Moist heat treatment is more likely to denature and coagulate protein, destroy protein chemical structure, thus destroying allergen epitope and reducing protein sensitization.

Reviewer 2 Report

This review is well-organized, easy to read, and appears to provide a comprehensive review of recent literature on walnut protein, with a focus on the global distribution. Some of the conclusions are straightforward and obvious, but others reflect the authors' presumably informed assessment of cases of seemingly contradictory information and the possible reasons for the variation. Overall, I think it would be a good starting point for anyone going into this area of research. A minor point: the manuscript is a Review, and the template is for an Article research.

Overall, this review provides a new sight into related society.

Author Response

Response to comments from reviewer 2:

  1. Some of the conclusions are straightforward and obvious, but others reflect the authors' presumably informed assessment of cases of seemingly contradictory information and the possible reasons for the variation. Overall, I think it would be a good starting point for anyone going into this area of research. A minor point: the manuscript is a Review, and the template is for an Article research.

Response: Suggestions were followed. Plant proteins has steered multiple attention in recent years, ascribing to its high nutrition and environmental friendliness. There are few reviews on walnut protein at present, and we hope to fill this gap through our efforts. The Article above the title has been changed to Review, as shown in line 1.

Reviewer 3 Report

Manuscript ID: foods- 2585864

Title: Advanced insights into walnut: A promising protein source for human health and food sustainability

Authors: Yuxuan Zhao, Weiheng He, Teng Jiao, Haifang Hu, Jingming Li, Lei Zhang, Jiachen Zang

Overview and general recommendation:

According to the authors the aim of the article is to provide a systematic review on walnut protein, with a focus on its physicochemical properties. Due to the great interest in alternative sources of protein, the topic of the manuscript is very interesting and topical, unfortunately, a lot of work will be required to redraft the manuscript so that it enriches the world of science with information about walnut proteins and does not duplicate common information about proteins - their production, structure and properties.

Major comments

1.          English language – first of all, I would recommend that the authors, after proofreading the manuscript, have it read by someone who knows scientific English well, as the article contains many colloquial expressions. The title itself suggests sensationalism, not a reliable analysis of data, and unfortunately this is the case throughout the manuscript. Examples from the text: “is obviously shown in Figure 1”, “Therefore, scholars from home and abroad”, “As new approaches…”

2.          My biggest concern about this review article is that it contains a lot of detailed information (too much) regarding knowledge about proteins in general - their structure, solubility, extraction and modification methods, including modern methods used to obtain various protein preparations - the problem is the fact that this is not information about walnut proteins, moreover, this information can be found in a very large number of generally available publications. And the authors, concluding the article, write: "This article has focused on the proteins from walnut…." - and about 70% of the manuscript does not even concern walnut proteins

3.           

Mainor comments

1.     Since this is a review article, it should not be classified as an "article" but a "review"

2.     I would also like to ask you to number the lines of the manuscript - it will make the work of reviewers easier and improve the process of preparing the manuscript.

3.     Tekxt

– “To meet the increasing food demand, the accelerated expansion of food manufacture triggers greater greenhouse gas (GHG) emissions, which is estimated to account for 35% of global total anthropogenic GHG emissions..” - please edit the sentence because it has no ending - conclusion, and the beginning of the sentence introduces the idea that there will be an ending - style error

-       “Juglans regia” - Latin names - in italics

-       Being authorized by FDA as a nutritional diet which may…” - Is it the whole diet, or is it an ingredient or compound, or can it be part of the diet?

-       “…and solve environmental and ecological problems” - this thought is probably too lofty, too few nuts are produced to solve such a global problem, it is certainly a more ecological alternative that may contribute to the fight for environmental protection, but it will not be a solution on its own.

-       systematic review on walnut protein” - what makes this article stand out from other scientific articles on the characteristics of walnuts? what new thing does it bring to literature?

-       I propose to separate the issues of the volume of walnut production in the world and the characteristics of the varieties and the quantity and quality of the ingredients contained in them, so that the information is complete and comparable, so that, for example, the quantity and quality of protein of the individual varieties mentioned in the article can be compared

-       Chapter 2.1 - Too detailed a description, the data can be presented in a table and the text should include only conclusions from these data and possibly what causes the differences in protein or protein and fat content, because for some varieties both values are given - maybe this information can also be unified?

-       “the amino acid types of Aksu walnut are complete” - please provide the correct wording amino acid composition not types, and clarify what complete means in case of walnut.

-       divided into four classifications” - whether these are four types of classification and not four types of proteins - this needs to be clarified

-       four independent proteins”; “independent four fractions” - what were those proteins? what does it mean independent? not walnut? need to clarify what is the point of doing it and putting these data in table 1

-       spectroscopy, there contained” – grammar mistake

-       Table 1. Amino acid composition of walnut proteins and protein fractions (g/100 g).” – if it is based on literature data- need to add the author, if it is own data need to add methodology

-       To better utilize the walnut proteins and its fractions, the understanding of their high structure is of key significance, while which is rarely reported up to now. 7S vicilin, a kind of typical globulins, is one of the rarely walnut proteins with crystal structure reported.” -references need to be add

-       its high structure by Rosetta (A comprehensive set of software for simulating macromolecular structures). The structure with highest estimated score, up to 0.938, was drawn in Figure 2c” - it might be worth providing the methodology of these calculations, as it is clear that they are an important source of information for the authors.

-       The emulsification stability of globulin is the highest,…” - the highest in relation to what?

-       3.2.3. Water and oil retention…” - Nothing specific follows from the information provided in the chapter, the properties of proteins are always influenced by the same parameters, such as pH, salt or temperature - but reading the article about walnuts, I expect that there should be something specific there - something that will allow us to compare the properties walnuts proteins to other, preferably widely known, so that this information can be used - this is definitely missing here.

-       When the temperature was 40℃, the maximum water holding capacity of walnut protein was 3.55.” – is it good or bad?

-       Sun et al. found that both of water and oil holding ability of walnut glutelin increased significantly.” - when they increased?

-       4.1. Walnut peptide” - highlight only the information that concerns walnuts

-       Curcumin characteristic – if the manuscript was about the characterization of walnut protein as a carrier in the microencapsulation of curcumin, it would make sense to characterize it deeply - in this case it is unnecessary

-       Figure 4. Utilization of walnut protein as a nanocarrier for carvacrol, with thermal stability and antibacterial activity significantly enhanced.” – in my opinion, the figure is unnecessary, it concerns a section of research, and the description is of several other studies on the topic

-       Figure 5. Schematic representation of walnut protein processing and products.” – and figures for other products - it would be worth approaching this comprehensively - there are many other products containing nuts, such as bread, meat products, etc. You can give up figures and describe it only in the text

-       Tree nut allergies affect an estimated 0.6% of the population of the United States [78], with English walnut (Juglans regia) as one of the most commonly reported allergenic tree nuts” - So what does this mean? Will a large percentage of people not be able to eat products with them?

-       Therefore, we need to improve…” – who is “we”?

-       Conclusions and prospects” - The conclusions are simply a summary, there is no conclusion here, whether for science or industry, what can the authors say based on the literature review about the potential of these walnut proteins?

English language – first of all, I would recommend that the authors, after proofreading the manuscript, have it read by someone who knows scientific English well, as the article contains many colloquial expressions. The title itself suggests sensationalism, not a reliable analysis of data, and unfortunately this is the case throughout the manuscript. Examples from the text: “is obviously shown in Figure 1”, “Therefore, scholars from home and abroad”, “As new approaches…”

Author Response

Response to comments from reviewer 3:

Major comments

  1. English language - first of all, I would recommend that the authors, after proofreading the manuscript, have it read by someone who knows scientific English well, as the article contains many colloquial expressions. The title itself suggests sensationalism, not a reliable analysis of data, and unfortunately this is the case throughout the manuscript. Examples from the text: “is obviously shown in Figure 1”, “Therefore, scholars from home and abroad”, “As new approaches…”

Response: Suggestions were followed. The title is changed to Advanced insights into walnut protein: Structure, physiochemical properties and applications. Change “is obviously shown in Figure 1” to “As shown in Figure 1” in line 88. Change “Therefore, scholars from home and abroad” to “Therefore, scholars are constantly exploring and improving the extraction process of walnut protein” in line 182. Change “As new approaches…” to “High-intensity ultrasound and high-pressure treatment has been approved” in line 192.

  1. My biggest concern about this review article is that it contains a lot of detailed information (too much) regarding knowledge about proteins in general - their structure, solubility, extraction and modification methods, including modern methods used to obtain various protein preparations - the problem is the fact that this is not information about walnut proteins, moreover, this information can be found in a very large number of generally available publications. And the authors, concluding the article, write: "This article has focused on the proteins from walnut…." - and about 70% of the manuscript does not even concern walnut proteins.

Response: Suggestions were followed. We deleted “Comparably, Dayao walnut, another classic walnut species in Yunnan province, owns higher protein content, reaching to 18.10%, which meets the China Walnut Quality Index (GB7907 -- 87)” in line 66, “Xiangling walnut in Shaanxi province is famous for its high protein content above 21.60%” in line 71, “The alkaline extraction-acid precipitation process usually results in an increase in the surface charge of extracted food proteins” in line 187, “As new approaches” in line 192, “According to Luo’s report, appropriate concentration of NaCl solution can improve the solubility and emulsification of protein meal” in line 213, “In addition, base extraction may lead to the formation of lysinoalanine, leading to potential toxicity and reducing the nutritional value of the synthesized protein” in line 461, “It can be used to modify food properties and homogenization, filtration, dehydration [94,95], emulsification, extraction, crystallization, depolymerization, fermentation, aging and microbial inactivation [96], which are considered to be important for protein extraction. High-energy mechanical waves (20-100 kHz) are conducted to induce cavitation and microfluidic currents. During acoustic processing, small bubbles form in the fluid and then collapse violently, creating localized regions of temperature (up to 5000 K) and high pressure (up to 1000 atm) around the collapsed cavity [97,98], thus leading to changes in the conformation of food proteins [99]” in line 467-476.

Mainor comments

  1. Since this is a review article, it should not be classified as an "article" but a "review"

Response: Suggestions were followed as shown in line 1.

  1. I would also like to ask you to number the lines of the manuscript - it will make the work of reviewers easier and improve the process of preparing the manuscript.

Response: Suggestions were followed.

  1. Tekxt

– “To meet the increasing food demand, the accelerated expansion of food manufacture triggers greater greenhouse gas (GHG) emissions, which is estimated to account for 35% of global total anthropogenic GHG emissions..” - please edit the sentence because it has no ending - conclusion, and the beginning of the sentence introduces the idea that there will be an ending - style error

Response: Suggestions were followed as shown in line 25. The accelerated expansion of food manufacture triggers great greenhouse gas (GHG) emissions.

- “Juglans regia” - Latin names - in italics

Response: Suggestions were followed as shown in line 28.

- “Being authorized by FDA as a nutritional diet which may…” - Is it the whole diet, or is it an ingredient or compound, or can it be part of the diet?

Response: It can be part of the diet.

- “…and solve environmental and ecological problems” - this thought is probably too lofty, too few nuts are produced to solve such a global problem, it is certainly a more ecological alternative that may contribute to the fight for environmental protection, but it will not be a solution on its own.

Response: We have changed the title to weaken the environmental roles of walnut protein. On the other hand, If walnut meal protein resources can be comprehensively utilized and its application scope broadened, it will be of great significance to the sustainable development of walnut industry and the development of health food.

- “systematic review on walnut protein” - what makes this article stand out from other scientific articles on the characteristics of walnuts? what new thing does it bring to literature?

Response: Thanks for your suggestion. This article discusses the value of walnut protein from the structural and physiochemical properties, functional attributes and some existing problems. Such reviews about walnut proteins have been rarely reported.

- I propose to separate the issues of the volume of walnut production in the world and the characteristics of the varieties and the quantity and quality of the ingredients contained in them, so that the information is complete and comparable, so that, for example, the quantity and quality of protein of the individual varieties mentioned in the article can be compared

Response: Thanks for your suggestion, and it is a good idea. In this review, the existing walnut protein data were sorted out mainly according to their location. But some detailed data about the quantity and quality of their ingredients have been rarely reported. We hope this review will attract more researcher to pay attention to walnut protein. It is also the researching work of ourselves.

- Chapter 2.1 - Too detailed a description, the data can be presented in a table and the text should include only conclusions from these data and possibly what causes the differences in protein or protein and fat content, because for some varieties both values are given - maybe this information can also be unified?

Response: Suggestion was followed. We have deleted some wordy data in Chapter 2.1.

- “the amino acid types of Aksu walnut are complete” - please provide the correct wording amino acid composition not types, and clarify what complete means in case of walnut.

Response: Suggestions were followed as shown in line 75. The average protein content of Aksu walnut cultivated in Xinjiang is 19.35%, and the essential amino acid content accounts for 27% ~ 30% of the total amino acid content, which meets the healthy demand of human beings.

- “divided into four classifications” - whether these are four types of classification and not four types of proteins - this needs to be clarified

Response: Suggestions were followed as shown in line 116. Walnut proteins can be divided into four types of proteins, namely water-soluble albumin, salt-soluble globulin, alcohol-soluble prolamins and alkaline-soluble glutens.

- “four independent proteins”; “independent four fractions” - what were those proteins? what does it mean independent? not walnut? need to clarify what is the point of doing it and putting these data in table 1

Response: Suggestions were followed as shown in 132. Walnut protein refers to the general term, while the four independent proteins refer to water-soluble albumin, salt-soluble globulin, alcohol-soluble prolamins and alkaline-soluble glutens.

- “spectroscopy, there contained” – grammar mistake

Response: Suggestions were followed as shown in 137-139. The analysis of circular binary chromatography (CD) shows that the secondary structure of walnut protein mainly includes α-helix, β-fold, β-rotation and random coil, among which α-helix and random coil are the most common.

- “Table 1. Amino acid composition of walnut proteins and protein fractions (g/100 g).” – if it is based on literature data- need to add the author, if it is own data need to add methodology

Response: Thanks for your suggestion, the issue can be answewed by the reference in line 129. Mao, X.Y.; Hua, Y.F.; Chen, G.G.  Amino acid composition, molecular weight distribution and gel electrophoresis of Walnut (Juglans regia L.) Proteins and Protein Fractionations. International Journal of Molecular Sciences. 2014, 15, 2003-2014. https://doi.org/10.3390/ijms15022003.

- “To better utilize the walnut proteins and its fractions, the understanding of their high structure is of key significance, while which is rarely reported up to now. 7S vicilin, a kind of typical globulins, is one of the rarely walnut proteins with crystal structure reported.” -references need to be add

Response: Suggestions were followed as shown in line 145. Zhang, M.L.; Gao, J.L.; Yang, H.X. Functional Properties of 7s Globulin Extracted from Cowpea Vicilins. Cereal Chemistry. 2009, 86, 261-266. https://doi.org/10.1094/CCHEM-86-3-0261.

- “its high structure by Rosetta (A comprehensive set of software for simulating macromolecular structures). The structure with highest estimated score, up to 0.938, was drawn in Figure 2c” - it might be worth providing the methodology of these calculations, as it is clear that they are an important source of information for the authors.

Response: Thanks for your suggestion. We attach a research paper by Zhang et al., where this measurement can be found. It is shown in line 145.

- “The emulsification stability of globulin is the highest,…” - the highest in relation to what?

Response: Thanks for your suggestion. The emulsification stability of globulin is the highest among all walnut proteins, which is helpful for its use as emulsifier in food processing. This sentence has been modified in line 210.

- “3.2.3. Water and oil retention…” - Nothing specific follows from the information provided in the chapter, the properties of proteins are always influenced by the same parameters, such as pH, salt or temperature - but reading the article about walnuts, I expect that there should be something specific there - something that will allow us to compare the properties walnuts proteins to other, preferably widely known, so that this information can be used - this is definitely missing here.

Response: Thanks for your suggestion. It is a common issue of all the plant proteins. After integrating the previous literature, the author also found that the properties of walnut protein need to be further analyzed. Thus in the conclusion, we put forward to pay more attention to the structural technologies of walnut protein as shown in line 508.

- “When the temperature was 40℃, the maximum water holding capacity of walnut protein was 3.55.” – is it good or bad?

Response: Suggestion was followed, we added more description in line 258. When the temperature was 40℃, the maximum water holding capacity of walnut protein was 3.55, much higher than before.

- “Sun et al. found that both of water and oil holding ability of walnut glutelin increased significantly.” - when they increased?

Response: Thanks for your suggestion. We added some details in line 268-274. The gluten in walnut cake protein was modified by enzymatic method and the microstructure of walnut protein was changed, thereby improving the functional properties of walnut protein. Sun et al. found that the modified walnut glutelin (WG) increased the solubility by ~1.33 times, the water holding capacity by ~0.23 times, the emulsifiability by ~0.32 times, and the emulsion stability by ~0.75 times. The oiliness was slightly lowered and the foaming characteristics were not greatly changed.

- Curcumin characteristic – if the manuscript was about the characterization of walnut protein as a carrier in the microencapsulation of curcumin, it would make sense to characterize it deeply - in this case it is unnecessary

Response: Thanks for your suggestion. We studied the characterization of walnut protein as carrier in microcapsules of curcumin in order to prove that walnut protein can be used as encapsulation material and has broad prospects.

- “Figure 4. Utilization of walnut protein as a nanocarrier for carvacrol, with thermal stability and antibacterial activity significantly enhanced.” – in my opinion, the figure is unnecessary, it concerns a section of research, and the description is of several other studies on the topic

Response: Suggestion was followed, Figure 4 was deleted in the revised manuscript.

- “Figure 5. Schematic representation of walnut protein processing and products.” – and figures for other products - it would be worth approaching this comprehensively - there are many other products containing nuts, such as bread, meat products, etc. You can give up figures and describe it only in the text

Response: Thanks for your suggestion. Figure 5 shows some of the commonly consumed products containing walnut protein, which the authors believe can clearly show the process from raw material to products. In order to clarify the question of the types of products containing walnut protein, we have added some details in the line 377.

- “Tree nut allergies affect an estimated 0.6% of the population of the United States [78], with English walnut (Juglans regia) as one of the most commonly reported allergenic tree nuts” - So what does this mean? Will a large percentage of people not be able to eat products with them?

Response: Thanks for your suggestion. Tree nut allergies affect 0.6 percent of the U.S. population, but this does not directly conclude that many people eat walnuts, only that there are allergenic ingredients in walnuts that are allergenic to Americans. This is why we ascribed “allergy” as a concern of walnut protein.

 “Therefore, we need to improve…” – who is “we”?

Response: Suggestions were followed as shown in 455. The main protein group (about 70%) of walnut is glutenin, and its poor water solubility limits its functional properties as a water-based food. Therefore, it is important to improve the physicochemical properties of walnut protein through various ways.

- “Conclusions and prospects” - The conclusions are simply a summary, there is no conclusion here, whether for science or industry, what can the authors say based on the literature review about the potential of these walnut proteins?

Response: Thanks for your suggestion, The pros and cons of walnut protein have been discussed in Parts 4 and 5, and it is undeniable that the walnut protein has great potential. We summarized the previous studies on walnut protein direction. Some future directions were also put forward in this part.

Reviewer 4 Report

Major comments:

P2, section2. Global distribution and characteristics of different species, The author describes data on walnut production based on 2018 data, but the reference FAO FAOSTAT data appears to be published 2021. I would prefer that the data used in this paper be updated to describe the most recent data. Also, the FAOSTAT reference states "Walnuts, in shell" rather than simply Walnuts, which I believe should also be accurately described. Based on the comments, Figure1 should be upgrade and insert what year data.

Considering the content of Section 2.1, it would be good to have a description of walnut varieties in the introduction. We believe that listing the number of major varieties, scientific names, and aliases, if any, will facilitate later understanding.

P3, Section2.2, I could not find the reference [8] and not follow the data based on ref.8.

Figure4, the caption and legend in the figure are slightly confusing. Although the abbreviations (WMPI, GA) are in the upper right corner along with the illustrations, I think these information should be placed directly on the page instead of the terms “WMPI” and “GA”. Also “carvacrol” in picture is difficult to see, it should be change the color or term place. 

Table2

According to Lyons SA (2021, https://doi.org/10.1016/j.jaip.2020.08.051), Jug r5 is not profilin, and profilin is Jug r7. Please update Table 2 information as the newest data. 

Minor comments:

Introduction section

L7, The term, Walnut scientific name “Jugulars regis L.” may be preferred Italic.

Line9, FDA should use the full spelling the first.

P2, Section 2.1, Jugulars sigillata => preferred italic.

P3, last line, pH 4.8–6.816. Probably(?), pH 4.8–6.8 [16]. Please check it out.

P4, regarding PDB, NCBI, spell out and then abbreviate. And the description of Rosetta software should be used when it first appears.

P5, section3.2.1. L7, Probably, “solubility around 90% 15” => “solubility around 90% [15].”

P7, section 4.1, 2nd paragraph L4, “the tripeptide leu-pro-phe (LPF)” => “the tripeptide Leu-Pro-Phe (LPF)”.

Also last 3 sentences, H2O2 subscribing and what is “EVSGPGLSPN”? Please explain the term.

Figure 3, Captions within figure 3 are slightly small. I think font size is larger.

P9L1, curucumin => CCM. Please show the reference about CCM bioavailability in stomach and intestine (p9, line2–4).

The data seemed a little out of date. It would be good to update the information as it should be up to date.

Author Response

Response to comments from reviewer 4:

Major comments:

  1. P2, section2. Global distribution and characteristics of different species, The author describes data on walnut production based on 2018 data, but the reference FAO FAOSTAT data appears to be published 2021. I would prefer that the data used in this paper be updated to describe the most recent data. Also, the FAOSTAT reference states "Walnuts, in shell" rather than simply Walnuts, which I believe should also be accurately described. Based on the comments, Figure1 should be upgrade and insert what year data.

Response: Thanks for your suggestion. The world production of walnut is classified as a matter of national security and trade secrets, so the data available is limited.

  1. Considering the content of Section 2.1, it would be good to have a description of walnut varieties in the introduction. We believe that listing the number of major varieties, scientific names, and aliases, if any, will facilitate later understanding.

Response: Thanks for your suggestion. We focused on walnut protein and revised the title. We also added some details in the line 58. Walnut belongs to the Juglandaceae family (Juglans) there are more than 20 species of walnut, distributed in Asia, Europe and the Americas (mainly North America).

  1. P3, Section2.2, I could not find the reference [8] and not follow the data based on ref.8.

Response:

Thanks for your suggestion. The website of reference 8 is attached below:

https://kns.cnki.net/KXReader/Detail?invoice=dmChCuGHSJlX85WsLxhgbrNxZWi8HdGSGdvcWHzz7MZNo6027kY86dRV7K%2BtDFLa5Y%2FFa7XCJhYfZiQXc8Wqkgn5ygGWFeCoPK0jzgp3L1IauUqEoI5DcljQ9hjYXTCs8WeZJJFm8NDN49d%2B7VpcQglrVXTUt1NV0pLR9uc9LMk%3D&DBCODE=CJFD&FileName=SXGS201604027&TABLEName=cjfdlast2016&nonce=E281A5848BDF424F923BB91DDD24F990&TIMESTAMP=1694345784844&uid=

  1. Figure4, the caption and legend in the figure are slightly confusing. Although the abbreviations (WMPI, GA) are in the upper right corner along with the illustrations, I think these information should be placed directly on the page instead of the terms “WMPI” and “GA”. Also “carvacrol” in picture is difficult to see, it should be change the color or term place.

Response: Suggestion was followed, Figure 4 was deleted in the revised manuscript.

  1. According to Lyons SA (2021, https://doi.org/10.1016/j.jaip.2020.08.051), Jug r5 is not profilin, and profilin is Jug r7. Please update Table 2 information as the newest data.

Response: Suggestions were followed as shown in Table 2.

Minor comments:

Introduction section

L7, The term, Walnut scientific name “Jugulars regis L.” may be preferred Italic.

Response: Suggestions were followed as shown in line 29.

Line9, FDA should use the full spelling the first.

Response: Suggestions were followed as shown in line 30.

P2, Section 2.1, Jugulars sigillata => preferred italic.

Response: Suggestions were followed.

P3, last line, pH 4.8–6.816. Probably(?), pH 4.8–6.8 [16]. Please check it out.

Response: Suggestions were followed as shown in line 136.

P4, regarding PDB, NCBI, spell out and then abbreviate. And the description of Rosetta software should be used when it first appears.

Response: Suggestions were followed as shown in line 145-157.

P5, section3.2.1. L7, Probably, “solubility around 90% 15” => “solubility around 90% [15].”

Response: Suggestions were followed as shown in line 178.

P7, section 4.1, 2nd paragraph L4, “the tripeptide leu-pro-phe (LPF)” => “the tripeptide Leu-Pro-Phe (LPF)”.

Response: Suggestions were followed as shown in line 293.

Also last 3 sentences, H2O2 subscribing and what is “EVSGPGLSPN”? Please explain the term.

Response: Suggestions were followed as shown in line 305.

Figure 3, Captions within figure 3 are slightly small. I think font size is larger.

Response: Suggestions were followed as shown in Figure 3.

P9L1, curucumin => CCM. Please show the reference about CCM bioavailability in stomach and intestine (p9, line2–4).

Response: Suggestions were followed as shown in line 349. Lv, S.Y.; Lu, Q.; Pan, S.Y. Stability and in Vitro Digestion of Pectin-Walnut Proteins Stabilized Emulsions Encapsulating Curcumin. Journal of Food Science. 2021, 8, 1-9.

https://kns.cnki.net/KCMS/detail/detail.aspx?dbname=cjfd2021&filename=spkx202108001&dbcode=cjfq

Round 2

Reviewer 3 Report

The authors improved the manuscript by adding fragments that directly concern walnuts, making the entire article more scientifically valuable. I think that with minor changes it can be further processed for publication.

Please add information in the table captions about where the data comes from, whether it is the authors' research or someone else did it. It is still not clear.

Reviewer 4 Report

Authors follow the reviewer's suggestions, but the format of the text is still flawed. Please check the references links and abbreviation format.